# Longitudinal Analysis and Latent Growth Modeling of the Modified Hip Dysfunction and Osteoarthritis Outcome Score for Joint Replacement (HOOS-JR)

**DOI:** 10.3390/healthcare12101024

**Published:** 2024-05-15

**Authors:** Emilie N. Miley, Michael A. Pickering, Scott W. Cheatham, Lindsay W. Larkins, Adam C. Cady, Russell T. Baker

**Affiliations:** 1Department of Orthopaedic Surgery and Sports Medicine, University of Florida, Gainesville, FL 32607, USA; emilie.miley1@gmail.com; 2Department of Movement Sciences, University of Idaho, Moscow, ID 83844, USA; michaela@tonypickering.com (M.A.P.); scheatham@uidaho.edu (S.W.C.); lindsayw.usc@gmail.com (L.W.L.); 3Kaiser Permanente, Woodland Hills, CA 91367, USA; adam.c.cady@kp.org; 4WWAMI Medical Education Program, University of Idaho, Moscow, ID 83844, USA; 5Idaho Office of Underserved and Rural Medical Research, University of Idaho, Moscow, ID 83844, USA

**Keywords:** total hip arthroplasty, invariance testing, latent growth curve modeling, patient-reported outcome measure

## Abstract

The Hip Dysfunction and Osteoarthritis Outcome Score for Joint Replacement (HOOS-JR) was developed as a short-form survey to measure progress after total hip arthroplasty (THA). However, the longitudinal validity of the scale structure pertaining to the modified five-item HOOS-JR has not been assessed. Therefore, the purpose of this study was to evaluate the structural validity, longitudinal invariance properties, and latent growth curve (LGC) modeling of the modified five-item HOOS-JR in a large multi-site sample of patients who underwent a THA. A longitudinal study was conducted using data from the Surgical Outcome System (SOS) database. Confirmatory factor analyses (CFAs) were conducted to assess the structural validity and longitudinal invariance across five time points. Additionally, LGC modeling was performed to assess the heterogeneity of the recovery patterns for different subgroups of patients. The resulting CFAs met most of the goodness-of-fit indices (CFI = 0.964–0.982; IFI = 0.965–0.986; SRMR = 0.021–0.035). Longitudinal analysis did not meet full invariance, exceeding the scalar invariance model (CFI_DIFF_ = 0.012; χ^2^_DIFF_ test = 702.67). Partial invariance requirements were met upon release of the intercept constraint associated with item five (CFI_DIFF_ test = 0.010; χ^2^_DIFF_ = 1073.83). The equal means model did not pass the recommended goodness-of-fit indices (CFI_DIFF_ = 0.133; χ^2^_DIFF_ = 3962.49). Scores significantly changed over time, with the highest scores identified preoperatively and the lowest scores identified at 2- and 3-years postoperatively. Upon conclusion, partial scalar invariance was identified within our model. We identified that patients self-report most improvements in their scores within 6 months postoperatively. Females reported more hip disability at preoperative time points and had faster improvement as measured by the scores of the modified five-item HOOS-JR.

## 1. Introduction

Total hip arthroplasties (THAs) are considered a highly effective treatment for patients who have been diagnosed with joint deterioration, such as osteoarthritis (OA), rheumatoid arthritis, and traumatic arthritis [1,2,3,4]. Nearly 300,000 THAs are performed annually in the United States, with rates predicted to grow by 174% from 2005 to 2030 due to increases in the population and disease diagnoses [2,5,6]. In addition, a large percentage (i.e., 70–85%) of THAs are performed on patients who have been diagnosed with arthritis, which is often associated with aging [4,7]. Not surprisingly, however, patients younger than 65 years of age have represented the fastest-growing age group for THAs, accounting for nearly 47% of all THAs performed in 2012 compared to only 34% in 1997 [2,5,6]. In addition to arthritis, other causes for patients undergoing a THA include developmental dysplasia of the hip and avascular necrosis, in patients younger than 65. However, current researchers have projected that more than 50% of all THAs will be performed on patients younger than 65 years of age by 2030 [6,8,9]. Thus, THAs are no longer only associated with the aging population [10].

Candidates for THAs often complain of pain and restricted range of motion (ROM) while performing activities [4,11]; as such, the goal of treatment is to reduce pain, improve ROM, and enhance overall quality of life (QOL) [1,2,4,10]. Many THA patients have failed to experience substantial improvement in their condition from conservative treatment approaches, which often leads to the decision to consider a THA [1,2,4,10]. Surgical and implant technologies have continued to evolve, producing excellent long-lasting outcomes that support THA as a treatment option in younger populations [7,10,12,13]. Patient-reported outcomes (PROs) are often used before and after THAs to measure pain, functionality, and QOL to evaluate surgical outcomes. The use of PROs helps clinicians and researchers determine the effectiveness of the treatment [13]. Due to the importance of outcomes to clinicians, researchers, and patients, it is imperative that PROs are valid, reliable, and responsive to change [13,14]; as such, PROs need to be evaluated over a prolonged period of time [15,16].

Outcomes currently used in the patient population with hip pain and dysfunction include the Hip Disability and Osteoarthritis Outcome Score (HOOS), and more recently, the Hip Dysfunction and Osteoarthritis Outcome Score for Joint Replacement (HOOS-JR) short-form survey [17,18,19,20,21,22,23,24]. The original one-factor, 6-item HOOS-JR was derived from the 40-item HOOS and developed as a short-form survey to measure outcomes specifically after surgical intervention (i.e., THA) and is primarily used pre- and postoperatively [25]. However, psychometric assessment of the HOOS-JR using contemporary methods has been minimally reported in the literature. Most clinometric assessments have focused on scale internal consistency, responsiveness of the instrument over three time points (i.e., preoperative, 6 months postoperative, and 12 months postoperative), and construct validity [25,26]. Currently, the six-item HOOS-JR has been reported to have acceptable internal consistency (0.86–0.87) and high responsiveness (0.80) [25]. Construct validity has previously been established for the six-item HOOS-JR by correlating it to the HOOS pain subscale (0.87), ADL subscale (0.94), symptoms subscale (0.62), sport and recreation subscale (0.65), and the QOL subscale (0.60). Also, the HOOS-JR was highly correlated to the Western Ontario & McMaster Universities Arthritis Index (WOMAC) function (0.94), pain (0.84), and stiffness (0.64) subscales [25]. The internal consistency of the HOOS-JR has also been established (Cronbach’s alpha = 0.86) [26]; values ranging 0.70-0.89 have been recommended to establish consistency without item redundancy [27,28]. 

Further research pertaining to scale validity (i.e., factor validity) on the HOOS-JR is needed [29]. This process should be performed to ensure the scale is appropriate for clinical practice and research [30,31]. To date, a limited number of studies have been conducted to assess the psychometrics of the six-item HOOS-JR using contemporary procedures, such as confirmatory factor analysis (CFA), exploratory factor analysis (EFA), or principal component factor analysis (PCA) methods [29,30,31]. Lyman et al. [25] performed a PCA to determine the dimensionality of the items; the authors identified that the 30 items, which were derived from the 40-item HOOS, were of a single construct. Further, this process, in combination with a Rasch analysis, reduced the number of items (N = 40) of the HOOS to form the six-item HOOS-JR [25]. 

In a previous study, the authors performed a CFA on the six-item HOOS-JR, identifying a promising factor structure which met most, but not all, of the model fit criteria (CFI = 0.965; TLI = 0.941; IFI = 0.965; RMSEA = 0.133) in a mostly healthy population [32]. Further, initial multi-group invariance testing was conducted; however, the analyses performed were only assessed at one time point and in a small sample (N = 656) [32]. In a more recent study, a CFA was performed on the six-item HOOS-JR in a large sample of patients who underwent a THA (N = 12,215) [33]. Goodness-of-fit indices were not met for the original six-item HOOS-JR (CFI = 0.938; TLI = 0.896; IFI = 0.938; RMSEA = 0.143); further exploration identified a more parsimonious scale structure (CFI = 0.981; TLI = 0.961; IFI = 0.981; RMSEA = 0.091) including five items of the original HOOS-JR (i.e., items two–six) [33]. As such, the modified five-item HOOS-JR warranted further investigation to confirm the psychometric properties pertaining to longitudinal use (i.e., outcome assessed over time).

Longitudinal research assessing the measurement properties of the modified five-item HOOS-JR over the course of treatment and long-term follow-up is also lacking. Valid instruments to assess change over prolonged periods of time are needed post-THA operation, because variations in outcomes may not emerge for years after the procedure [15,16]. Establishing measurement properties through invariance testing ensures that the interpretations across time are valid and reliable [27,34], and longitudinal measurement invariance assessment is conducted to determine if individual responses at each time point are representing a similar underlying construct, which is necessary to assess changes over time accurately [30,35]. In addition, assessing the longitudinal data of the five-item HOOS-JR may is important for clinicians to understand how patients perceive changes in pain and level of function during the healing process. Latent growth curve (LGC) modeling is a statistical technique designed to study the change in data when the outcome variable (e.g., five-item HOOS-JR) is collected at multiple time points [35,36,37,38]. More recently, researchers have performed LGC modeling to assess changes in PRO scores (i.e., Oxford Hip Score [OHS] and Harris Hip Score [HHS]) pre- and postoperatively and noted that most of the healing occurs within the first 3 to 6 months postoperatively [39,40]. Thus, further measurement research on the modified five-item HOOS-JR is needed to establish scale validity over time for its clinical use, to determine the rates of which healing is measured through the responses of the five-item HOOS-JR, and also determine if differences exist in these rates of healing patterns based on age and sex. 

As studies have failed to support the use of the six-item HOOS-JR and identified a new five-item HOOS-JR, longitudinal assessment of the proposed scale is warranted. In addition, studies assessing LCG modeling or assessing if differences exist based on age and sex of any forms of the HOOS-JR are lacking in the literature. Therefore, the purpose of this study was to evaluate the psychometric properties of the modified five-item HOOS-JR in a three-step process: (1) a CFA of the factor structure using contemporary fit recommendations in a large sample of patients who underwent a THA to ensure model fit at each time point (i.e., preoperatively and 6 months, 1 year, 2 years, and 3 years postoperatively); (2) longitudinal invariance testing (i.e., equal factor variances, equal factor covariance, and equal means) of the scale across multiple time points (i.e., preoperatively, and 6 months, 1 year, 2 years, and 3 years postoperatively); and (3) LCG modeling to assess the heterogeneity of the recovery patterns for different subgroups of patients (i.e., age groups and sex).

## 2. Materials and Methods

### 2.1. Data Source and Participants

The Surgical Outcomes System (SOS; Arthrex, Inc., Naples, FL, USA) database was retrospectively queried between the years 2014 and 2020 to establish a large dataset for assessment. The Institutional Review Board (IRB) was not required for this study as the SOS is an international registry of de-identified data that adheres to the Health Insurance Portability and Accountability Act (HIPPA) regulations. The SOS contains information submitted from multiple orthopedic surgical centers around the world. Specifically for this study, patients were queried who underwent a THA procedure and completed the 6-item HOOS-JR. Demographic information (i.e., sex, age at treatment) and responses to the 6-item HOOS-JR were extracted from the SOS database for analysis.

### 2.2. Hip Dysfunction and Osteoarthritis Outcome Score for Joint Replacement

The modified HOOS-JR includes five items that ask participants to rate how frequently they engaged in behaviors over the past week using a 5-point Likert scale (0 = none, 1 = mild, 2 = moderate, 3 = severe, and 4 = extreme) [25]. The five items included were items two through six of the original 6-item HOOS-JR. To be consistent with the original HOOS-JR, scores of the 5-item HOOS-JR were calculated by creating a raw sum score of the five items (i.e., items two through six of the original scale). These methods are similar to the calculation guidelines presented by Lyman et al. [41]; however, no interval score was calculated. Therefore, the highest score (i.e., 20) indicated total hip disability and the lowest score (i.e., 0) indicated perfect hip health.

### 2.3. Data Analysis

#### 2.3.1. Data Cleaning

Using Microsoft^®^ Excel for Mac (Version 16.46; Redmon, WA, USA), data were exported from the SOS database and uploaded to Statistical Package for Social Sciences Version 24.0 (IBM Corp., Armonk, NY, USA) for data cleaning and analysis purposes. Because the primary purpose was to assess the HOOS-JR, individuals with missing demographic data were not excluded from analysis and were left as missing values. Univariate outliers were assessed using z-scores, and participants with z-scores that exceeded the cut-off value of |3.3| were removed [30]. The presence of multivariate outliers was also assessed; participants were assessed, flagged, and removed if the Mahalanobis distance, using a chi-square table with degrees of freedom and *p*-value = 0.01 [30,42], was exceeded [30]. For longitudinal invariance, participants who did not respond to the HOOS-JR items at all five time points were not used in the analysis.

#### 2.3.2. Confirmatory Factor Analysis

The full sample was used to conduct a CFA to assess the scale structure of the 5-item HOOS-JR using Analysis of Moment Structures (AMOS) software (Version 27; IBM Corp., Armonk, NY, USA) at each time point (i.e., preoperatively, and 6 months, 1 year, 2 years, and 3 years postoperatively). The modified HOOS-JR was specified as a one-factor, 5-item model to remain consistent with our previous work. Full Information Maximum Likelihood estimation was used to generate the parameter estimates [30,43]; model fit statistics were assessed based on a priori values. Goodness-of-fit indices used included the likelihood ratio statistic (CMIN), Goodness-of-Fit Index (GFI; ≥0.95), Comparative Fit Index (CFI; ≥0.95), Tucker–Lewis Index (TLI; ≥0.95), Bollen’s Incremental Fit Index (IFI; ≥0.95), Root Mean Square Error of Approximation (RMSEA; ≤0.05), and Standardized Root Mean Square Residual (SRMR; ≤0.08) [30,43,44]. The interpretability, size, and significance of the model’s parameter estimates (i.e., factor variances, covariances, and indicator errors) were examined to identify any localized areas of strain in addition to assessing the overall goodness-of-fit [45]. 

#### 2.3.3. Longitudinal Invariance Testing

The same criteria utilized for the CFAs were used to assess fit for the invariance model [30,45]. Invariance testing was conducted with the full sample to assess measurement invariance of the 5-item HOOS-JR across five time points (i.e., longitudinal invariance). Longitudinal invariance testing assesses the model fit of all time points simultaneously, while the prior step (i.e., assessing the scale structure using CFA) tests each time point individually [30,45]. Therefore, individuals who completed the 6-item HOOS-JR at all five time points (i.e., preoperatively, and 6 months, 1 year, 2 years, and 3 years postoperatively) were used to assess invariance across time. A CFI difference (CFI_DIFF_) of less than 0.01, and a chi-square (χ^2^_DIFF_) with a *p*-value cut-off of 0.01, was evaluated for structural invariance [30,45]. As the χ^2^_DIFF_ test is sensitive to sample size, the CFI_DIFF_ test held a greater influence in decisions regarding invariance testing [30,45]. If a model passed the CFI_DIFF_ test, but exceeded the χ^2^_DIFF_ test, invariance testing would continue.

Longitudinal invariance implies that patients, across repeated measures, interpret the questions and construct (i.e., hip disability) in the same way [30,45]. If a model is invariant, comparisons across repeated measures are possible which can provide clinicians with the ability to assess change over time and conclude that the measured change was true and not due to measurement error. As such, configural, metric, and scalar invariance testing was performed to assess the scale structure across five time points [30,43,45]. In the configural invariance model, the latent structure was constrained to be equal across time. Secondly, in the metric invariance model, additional constraints were placed on the item loadings across time [43,45]. If the model met metric invariance requirements, equal variances (i.e., differences in scores at different time points) across time were then assessed [43,45]. Lastly, in the scalar invariance model, the latent structure, item loadings, and item intercepts were set to be constrained (i.e., equal) across time [43,45]. If the model met scalar invariance requirements, equal mean models (i.e., score differences) were tested across time points [43]. If model invariance was not found, investigation of the source leading to non-invariance was explored by sequentially releasing item loadings or intercepts to explore if a partially invariant model could be identified [46]. 

#### 2.3.4. Longitudinal Invariance Testing

As the model fit of the modified 5-item HOOS held during longitudinal invariance testing, the sample was then subjected to LGC modeling using AMOS [30]. The LGC model analysis allows for the evaluation of model adequacy using model fit indices and model selection criteria, which accounts for measurement error using latent repeated measures [38,44]. As such, the LGC modeling was conducted on the modified 5-item HOOS-JR using Full Information Maximum Likelihood, as this method is best for handing missing data [30]. First, a priori values were hypothesized to be linear [43], meaning a decrease in pain and an increase in function from preoperative to postoperative time points. The same goodness-of-fit criteria utilized for the CFAs were used to assess fit for the LGC model [30,35,43,45]. A growth trajectory (i.e., intercept and slope) was assessed for intraindividual and interindividual differences to determine the direction and extent to which the patient’s self-perceived hip disability changed from preoperative to 3 years postoperatively. The intercept factor represents the starting point (i.e., preoperative) for the trajectory of a factor (i.e., 5-item HOOS-JR), whereas the slope represents the change in the trajectory of the 5-item HOOS-JR over time [47]. 

The model included two growth parameters: (a) the intercept parameter, representing an individual’s score on the HOOS-JR at the preoperative time point (time point 1), and (b) the slope parameter, representing the individual’s rate of change over the 3-year follow-up [43]. Also, covariances between the intercept and slope factors were also included; this provides an indication of whether patients who started at a lower or higher score for the outcome changed at lower or higher rate [43]. Values assigned to the slope parameters were represented as preoperative = 0, 6 months postoperative = 0.5, 1 year postoperative = 1, 2 years postoperative = 2, and 3 years postoperative = 3 [43,48]. If the interindividual growth trajectory was statistically significant (i.e., the sample was heterogeneous), multi-group testing was then conducted to determine if the sample could be further explained [30,43]. Groups of interest extracted from the database included preoperative patient characteristics including age groups (i.e., <45, 45–54, 55–64, 65–74, ≥75) and sex (i.e., male, female). If a nonlinear growth model was found, slope parameters were freely estimated to explore the shape of the growth model [43,48]. 

## 3. Results

A total of 1707 complete participant responses (i.e., all items of the original HOOS-JR answered at all five time points) were extracted for data cleaning. From the total sample with responses at all five time points, univariate (N = 0; 0.0%) and multivariate (N = 40, 2.3%) outliers were identified at the preoperative stage and were removed during the data cleaning process. Deleted cases included females (N = 23) and males (N = 14), with an average age of 65.05 ± 10.01 y (range = 46–86). The final sample (N = 1667) used for analysis consisted of 48.8% males (N = 769) and 51.2% females (N = 806), with a mean age of 61.72 ± 9.90 y (range = 24–90 y). Of the final sample of responses, 5.5% (N = 92) were missing a response to sex.

### 3.1. Confirmatory Factor Analysis

The CFA of the modified five-item HOOS-JR at each time point (e.g., preoperatively, 6 months postoperatively) indicated an acceptable fit of the data (Table 1) [30,43,44]. Model fit indices met the recommended values for the CFI (range = 0.964–0.982), IFI (range = 0.965–0.986), and SRMR (0.021–0.035). However, TLI (range = 0.929–0.972) and RMSEA (range =0.081–0.126) values slightly exceeded recommendations [30,43,44]. Factor loadings were significantly different (*p*-values < 0.001) and ranged from 0.68–0.82. At time points 3 and 4, modification indices revealed meaningful high error-term covariances between item five (i.e., lying in bed [turning over, maintaining hip position]) and item six (i.e., sitting). However, no cross-loadings were identified between items at any time point.

### 3.2. Longitudinal Invariance Testing of the Alternate HOOS-JR

Analysis of the five time points (i.e., preoperatively, and 6 months, 1 year, 2 years, and 3 years postoperatively) revealed the initial model (i.e., equal form) met all model fit indices (CFI = 0.975; χ^2^ = 804.79; RMSEA = 0.040; Table 2) [30,43,44]. The metric model (i.e., equal loadings) passed both the CFI_DIFF_ test (CFI = 0.969) and the χ^2^_DIFF_ test (χ^2^ = 145.94) [30,43,44]. Because the metric model was invariant between time points, an examination of the equal latent variable factors was warranted. The equal factor variance model slightly exceeded the CFI_DIFF_ test (CFI = 0.964) and the χ^2^_DIFF_ test (χ^2^ = 286.66), indicating the variances of the latent variables were not equal across time points [30,43,44]. The scalar model (i.e., equal intercepts) also slightly exceeded the CFI_DIFF_ test (CFI = 0.946) and the χ^2^_DIFF_ test (χ^2^ = 702.67), suggesting item-level bias [30,43,44]. However, because the scalar model only slightly exceeded the CFI_DIFF_ (i.e., 0.012) and χ^2^_DIFF_ tests, the evaluation of the equal latent means model continued. Upon assessment of the equal latent means, the model did not pass either the CFI_DIFF_ test (CFI = 0.835) or the χ^2^_DIFF_ test (χ^2^ = 3323.33). When the means were not constrained to be equal across time points, significant mean score improvement was found between time point 1 and time point 2 (i.e., −1.69), time point 2 and time point 3 (−1.75), and time point 3 and time point 4 (i.e., −1.83). Mean scores remained consistent (i.e., −1.83) from time point 4 to time point 5, indicating similar mean scores between the two time points that were lower (i.e., indicating improved hip health maintained across time) when compared to preoperative, 6 months postoperative, and 1-year postoperative mean scores.

Because the model did not meet the strictest criteria for scalar invariance, partial invariance at the scalar level was explored by sequentially releasing intercepts of the items [30,43,46]. Item five (i.e., lying in bed [turning over, maintaining hip position]) was identified as the source of non-invariance; when released, the scalar model (i.e., equal intercepts) met the CFI_DIFF_ test (CFI = 0.965) and the χ^2^_DIFF_ test (χ^2^ = 1073.83; Table 3) [30,43,44]. Because the scalar passed the CFI_DIFF_ and the χ^2^_DIFF_ test, the evaluation of the equal latent means model continued. Upon assessment of the equal latent means, the model did not pass either the CFI_DIFF_ test (CFI = 0.842) or the χ^2^_DIFF_ test (χ^2^ = 3962.49; Table 3) [30,43,44]. When the means were not constrained to be equal across time points, the mean scores improved between time point 1 and time point 2 (−1.66), time point 2 and time point 3 (−1.76), and time point 3 and time point 4 (−1.84) and remained consistent between time point 4 and time point 5 (−1.84).

### 3.3. Longitudinal Invariance Testing of the Alternate HOOS-JR

The linear LGC model did not meet the recommended model fit indices (CFI = 0.049, TLI = 0.049, IFI = 0.049, RMSEA = 0.424; Figure 1) [43]. These findings suggest that patients’ responses to the questionnaire were nonlinear. Therefore, exploratory methods were used to identify the shape of the growth trajectory. 

Slope parameters for time points 1 and 2 were constrained to 0.0 and 0.50, respectively; however, the remainder of the time points (i.e., 3–5) were freely estimated [43,48]. Upon assessment of these findings, the final slope parameters were defined as follows: preoperative = 0.0, 6 months postoperative = 0.92, 1 year postoperative = 0.96, 2 years postoperative = 1.0, and 3 years postoperative = 1.0. This model met the recommended model fit indices (CFI = 0.986; TLI = 0.986; IFI = 0.986; RMSEA = 0.051; Figure 2) [30,43,44]. Upon assessment of the means, the estimates pertaining to the intercept and shape were statistically significant (*p* < 0.001). In addition, our findings revealed that the average mean score for modified five-item HOOS-JR was 10.48 points at the preoperative time point and decreased over three years (i.e., −8.48 points). When assessing the covariance between the intercept and shape, a negative estimate was identified (i.e., −2.08; Figure 2). Variances of the model estimates were not significantly different for the intercept (*p* = 0.06); however, they were statistically different for the shape (i.e., *p* = 0.05).

Model fit indices were also assessed between groups (i.e., sex and age) to determine the differences that sex and age groups had on the mean scores and growth trajectories over time. When assessing the LGC model pertaining to sex, the model met all recommended fit indices (CFI = 0.981, TLI = 0.981, IFI = 0.981, RMSEA = 0.043) [30,43,44]. Upon assessment of the estimates pertaining to the means, the parameters for both the intercept and the shape were statistically significant (*p* < 0.05). Upon assessment of the mean scores, the average mean score for the modified five-item HOOS-JR at preoperative visit were higher in patients within the female group (10.92) compared to patients in the male group (10.01). In addition, patients in both groups improved their scores over the 3 years, with patients in the female group having an overall greater decrease in scores (i.e., females = −8.98, males = −7.99). When assessing the covariance between the intercept and shape, a negative estimate was identified for patients in the male group (i.e., −3.72); however, the female group had a small, yet positive change (i.e., 0.32). Lastly, when assessing the variances of the model, estimates related to the intercept (*p* = 0.050) and shape (*p* = 0.049) were significant.

When assessing the LGC model for age groups, the model met all recommended fit indices (CFI = 0.979, TLI = 0.979, IFI = 0.973, RMSEA = 0.028) [30,43,44]. On average, patients in the 65–74 age group had an overall worst score at the preoperative visit (i.e., 11.21) when compared to patients in the other age groups. In addition, patients in all groups increased their scores over the 3 years, with patients in the <45 age group having the greatest decrease in scores (i.e., −9.30). When assessing the covariance between the intercept and shape, those aged <45 had the largest negative estimate (−13.09) compared to the other age groups. Lastly, when assessing the variances of the model, estimates related to the intercept and shape were not significant (*p* = 0.185, *p* = 0.113, respectively).

## 4. Discussion

Previous research on the original six-item HOOS-JR did not meet the recommended goodness-of-fit indices pertaining to structural validity [33]; therefore, further assessment of the modified five-item HOOS-JR was recommended. The purpose of our study was threefold: (1) assess the structural properties, (2) assess the longitudinal invariance properties, and (3) assess the LGC model characteristics (i.e., rate of perceive change in scores) of the modified five-item HOOS-JR in a sample of patients who underwent a THA and completed the original HOOS-JR items at multiple visits (i.e., preoperatively, and 6 months, 1 year, 2 years, and 3 years postoperatively). In addition, we sought to assess the heterogeneity in the responses to the five-item HOOS-JR across subgroups (i.e., age groups and sex) using LGC modeling to determine if differences existed in recovery following a THA. Our results indicate that the modified five-item HOOS-JR can be used across different time points. 

However, caution is warranted when attempting to make comparisons using the scores across time. As concerns pertaining to the scale (i.e., high error-term covariances and lack of full measurement invariance) still exist, consideration to implement an instrument that better captures the full scope of recovery pertaining to the hip joint is warranted. We did not identify significant group differences between age groups or sex, indicating that regardless of the group (i.e., different age groups or males vs. females), patients answered the modified five-item HOOS-JR similarly. Lastly, most of the improvements in scores (i.e., patients reporting less hip disability) occurred within the first 6 months to 1 year postoperatively.

### 4.1. Confirmatory Factor Analysis

The psychometric properties pertaining to the structural validity of the modified five-item HOOS-JR were assessed individually at five time points (i.e., preoperatively, and 6 months, 1-year, 2 years, and 3-postoperatively). Most model fit indices were met at each time point, indicating that the modified five-item HOOS-JR had suitable structural validity for measuring hip disability at the five time points [30,34,35]. However, upon further assessment, time point 3 (i.e., 1 year postoperatively) and time point 4 (i.e., 2 years postoperatively) indicated meaningful high error-term covariances between items five (i.e., lying in bed [turning over, maintaining hip position]) and item six (i.e., sitting). Meaningful error-term covariances indicate a possible model misspecification, which may demonstrate the need for further refinement of the scale [45]. In addition, we found similar findings within our previous work when assessing scale validity in a larger sample of patients (N = 12,215) at one time point (i.e., preoperatively); meaningful high error-term covariances were also present between these two items. The overall findings support the need for further refinement of the items, such as re-writing or removing items, to produce a more parsimonious and psychometrically sound instrument [30,43]. 

Assessment of the RSMEA value revealed a higher-than-recommended cut-off value (i.e., ≤0.05) at each time point. In our previous work, we identified similar RMSEA findings (i.e., 0.143) [30,43]. However, due to the small df at each individual time point (i.e., 5), we assessed the SRMR in conjunction with the RMSEA, as previous studies report that models with small df can negatively influence the RMSEA and cause potential model misfit [49,50]. As such, authors recommend reporting the SRMR in models containing small df [49,50]; therefore, SRMR and CFI combined held greater weight in our decision regarding model fit interpretation [49]. With this in mind, SRMR fit indices were met at all time points.

### 4.2. Longitudinal Invariance Testing

Longitudinal invariance was conducted to determine if the modified five-item HOOS-JR was structurally invariant across multiple time points. Longitudinal invariance was identified for the configural (equal form) and metric (equal loadings) model, indicating that patients interpreted the questions and underlying construct (i.e., hip disability) in the same way across repeated visits [30,43]. However, the scalar model (equal intercepts) slightly exceeded the strictest invariance criterion; therefore, subsequent analyses were performed to determine the item intercept that was the source of invariance [43,46]. The results of these analyses revealed that item five (i.e., lying in bed [turning over, maintaining hip position]) was not interpreted similarly by patients across the repeated visits. When the intercept associated with item five was not constrained, the model passed scalar invariance. These findings allow clinicians to be able to assess change over the repeated visits and allow us to conclude that the change measured was not due to measurement error, but true change in the patients’ perceived hip disability. As scalar invariance was identified with the release of item five, caution is warranted clinically when incorporating the score of this item in the assessment of mean scores over time [30,46]. 

The invariant solution allowed us to assess the equal factor variances and equal means model. The equal latent variances model slightly exceeded the χ^2^_DIFF_ test and the CFI_DIFF_ difference test, indicating that variances were not equal across time. When the variances were not constrained to be equal, there was more variability (i.e., 0.49) pertaining to hip disability at the preoperative visit compared to the postoperative time points (range = 0.25–0.30), indicating that the variances in scores decreased over time [30,34,35]. Upon assessment of the equal latent means model, the χ^2^_DIFF_ test and the CFI_DIFF_ difference test were exceeded [30,34,35]. When the means were not constrained to be equal, patients responded with the highest score (i.e., more hip disability) preoperatively, and the lowest scores (i.e., less hip disability) at 2 and 3 years postoperatively. However, the majority of improvement in scores happened between the preoperative time point and 6 months postoperatively; these findings are similar to previous researchers, who noted the most improvement in HOOS-JR scores within the first 6 months post-THA operation [39,40,51,52]. 

### 4.3. Latent Growth Curve Model

Minimal research has focused on LGC modeling pertaining to outcomes related to THA [39,40,51]. Prior studies pertaining to the OHS [39,40], HOOS Physical Function (PS) [51], and HHS [52] have included LGC modeling. To our knowledge, this was the first study to assess LGC modeling in patients who answered questions to the HOOS-JR over any longitudinal period, and between age and sex groups following THA. Utilizing LGC modeling is unique because it allows researchers to assess individual growth and interindividual differences in change longitudinally [43,44,53]. 

Our LGC model did not meet the recommended model fit indices pertaining to a purely linear model (i.e., constant rate of change over time). However, when the intercepts were constrained to be nonlinear, the model met the recommended model fit indices. These findings suggest that patients improve their hip disability, as measured by the modified five-item HOOS-JR, at a nonlinear rate. With the use of LGC modeling, we identified significant differences for the intercept and shape trajectory (*p* < 0.001). Overall, patients’ average mean scores on the modified five-item HOOS-JR were 10.48 and decreased by 8.48 points from preoperative to 3 years postoperative. These findings suggest that patients decrease their scores on the five-item HOOS-JR over time (i.e., improved hip disability) following a THA. Within our model, most score improvement occurred within the first 6 months postoperatively. Also, when assessing the covariance between intercept and shape trajectory, a negative value was identified (−2.08), indicating that patients who reported less hip disability at the preoperative visit demonstrated a lower rate of improvement in HOOS-JR scores over the 3 years postoperatively [30,43]. These findings are consistent with prior THA research that identified the largest improvement in PRO scores (i.e., OHS, HHS, HOOS-PS) within the first 3 to 6 months postoperatively [39,40,51,52]. In addition, changes in the mean scores of the modified five-item HOOS-JR were similar to other studies assessing the differences in PRO scores over time [39,40,51,52].

We also assessed between-group (i.e., age groups and sex) differences to test for differences between sex and age groups across time. When assessing the means (i.e., intraindividual differences), the intercept and shape trajectory were statistically significant (*p* < 0.001). These findings reveal that the average scores of the modified five-item HOOS-JR notably differ; average scores were lower for males (i.e., 10.01) compared to females (10.92). As such, patients in the female group reported greater hip disability compared to the male patients at preoperative visits. Our findings are consistent with the previous literature that indicated that females reported higher levels of hip disability than males prior to surgery [54,55]. In addition, patients in both the male and female groups demonstrated a decrease (i.e., improvement) in average scores (−7.99 and −8.98, respectively) from preoperatively to 3 years postoperatively. 

When assessing the covariances between intercept and slope, females had a faster change in mean scores (i.e., 0.32) compared to males (i.e., −3.73); however, the negative covariance represents patients who scored lower at the preoperative time point (i.e., less hip disability), demonstrating a lower rate of increase in scores over the 3-year period. Lastly, when assessing the variances of the model (i.e., interindividual differences), estimates related to the intercept (*p* = 0.050) and shape (*p* = 0.049) were significant. These findings reveal that there are significant differences in the variances of the mean scores at the preoperative visit and over time associated with the modified five-item HOOS-JR. To our knowledge, this was the first study assessing the variances in scores between males and females longitudinally (i.e., over time) from preoperatively to 3 years postoperatively. In a previous study, the authors reported the mean score differences at preoperative and 5 years postoperatively [52]. Even though the authors found similar findings at the preoperative visit (i.e., significant differences between males and females), the amount of hip disability was not significantly different between sexes at the 5-year postoperative visit [52]. 

In addition, we sought to determine if differences in scores existed between age groups (i.e., <45, 45–54, 55–64, 65–74, ≥75). Our LGC model met all recommended fit indices (CFI = 0.98, TLI = 0.978, IFI = 0.980, RMSEA = 0.029). When assessing the means, the intercept and the shape trajectory were statistically significant (*p* < 0.001); patients in the <45 age group reported significantly higher scores of the modified five-item HOOS-JR compared to patients within the other age groups. In addition, patients in the 65–74 age group reported the lowest average score (i.e., better hip health) on the modified five-item HOOS-JR scores (10.08) preoperatively. However, the average scores significantly decreased the most for patients within the <45 age groups (−9.30), whereas patients in the 65–74 age group decreased the least (−8.15). Our findings reveal that those who are younger reported greater improvement in their hip disability from preoperative to 3 years postoperatively compared to other age groups [43]. Our findings support prior research that demonstrates that younger patients report faster recovery patterns following a THA when compared to older patients [40]. 

Lastly, we identified a larger negative covariance between intercept and slope (i.e., −13.09) in patients in the <45 age group. These findings reveal that patients <45 who reported lower scores (i.e., less hip disability) of the modified five-item HOOS preoperatively demonstrate a lower rate of improvement in scores over the 3 years postoperatively; whereas patients in the 55–64 age group had a larger, positive (i.e., 2.89) covariance, suggesting patients have a faster rate of improvement in hip disability as measured by the modified five-item HOOS-JR scores [35,48]. Of note, however, even though we identified intraindividual differences in average mean scores between age groups, these findings were not statistically different between groups [35,48]. When assessing interindividual differences in both the initial scores and their change over time (i.e., variances of the model), estimates pertaining to the intercept (*p* = 0.185) and slope (*p* = 0.113) were not significant [35,48].

### 4.4. Limitations and Future Research

Even though this study is the first to our knowledge to explore longitudinal invariance testing and LGC modeling of the HOOS-JR, several limitations were present that warranted discussion. Even though we had a large sample of patients who completed the 6-item HOOS-JR at five different time points, further research should include a confirmation sample to ensure similar findings exist for the modified five-item HOOS-JR. Our sample lacked complete demographic information pertaining to the dataset, limiting further understanding of the population assessed. In addition, the lack of demographic information restricted the analyses of additional subgroups (e.g., surgical approach, ethnicity, race, socioeconomic status). Future research should include more demographic information for further testing. In addition, clinicians should practice caution using the five-item HOOS-JR in clinical practice and throughout postoperative care when examining group differences in scores of the modified five-item HOOS-JR in subgroups not yet analyzed. We performed longitudinal invariance testing and LGC modeling on the modified five-item HOOS-JR which was identified in our previous work. However, the sample of patients responded to the six-item HOOS-JR. It is possible that responses were influenced by the additional item present on the original six-item HOOS-JR scale. Future research should be conducted on patients who only respond to the items on the modified scale. We found item five to be problematic (i.e., high error-term covariance); as such, partial invariance was identified for the structure by removing the intercept constraint associated with the item during assessment of the scalar model. Therefore, further scale modifications (e.g., re-wording or removal of the item) may be pertinent and caution is recommended when assessing the change in mean scores of the five-item HOOS-JR across time.

## 5. Conclusions

To our knowledge, this was the first study to assess longitudinal invariance testing and perform a multi-group LGC modeling approach for the modified five-item HOOS-JR. As the five-item HOOS-JR did not meet the strictest longitudinal measurement invariance criteria (i.e., full scalar invariance), partial scalar invariance was identified by releasing the scores pertaining to item five. In addition, we identified that patients self-reported most improvement in their scores within the first 6 months postoperatively. We found significant differences in mean scores between sex, with females reporting more hip disability preoperatively and reporting a faster improvement as measured by the scores of the modified five-item HOOS-JR. Researchers and clinicians can use the scale at different time points; however, caution is warranted when attempting to compare mean scores across time.

## Figures and Tables

**Figure 1 healthcare-12-01024-f001:**
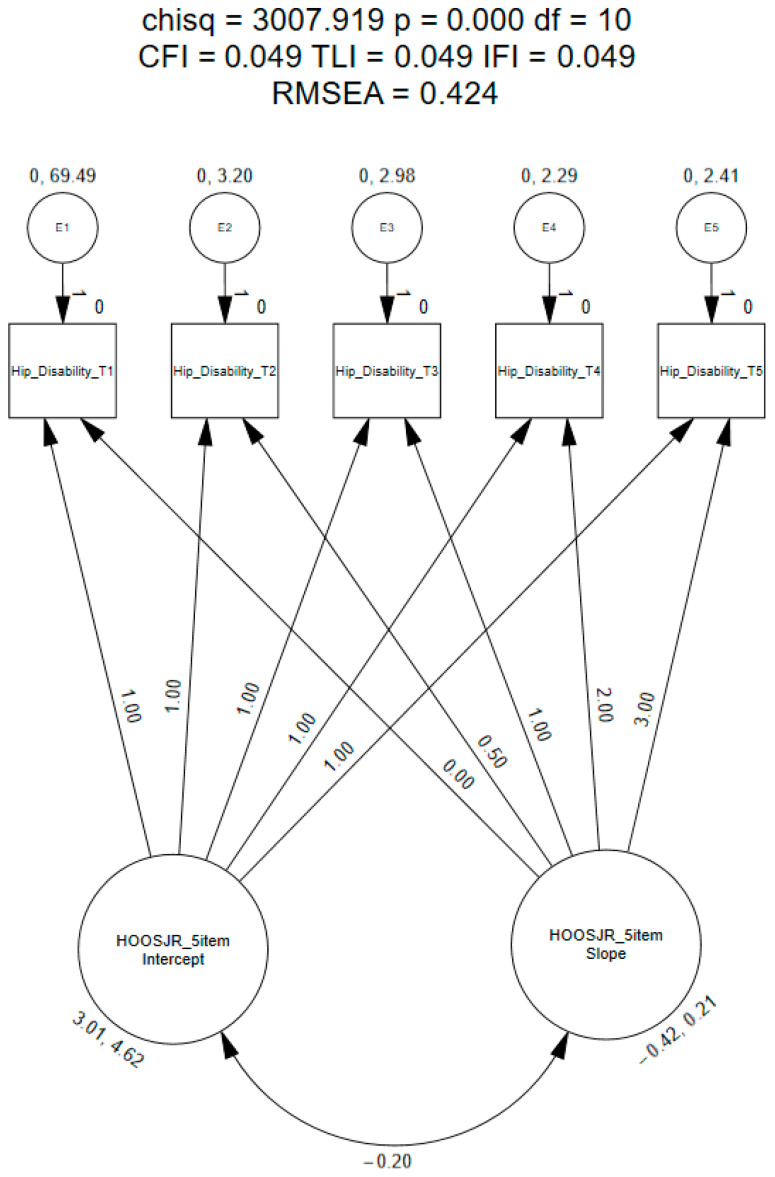
Linear latent growth curve model of the modified five-item HOOS-JR.

**Figure 2 healthcare-12-01024-f002:**
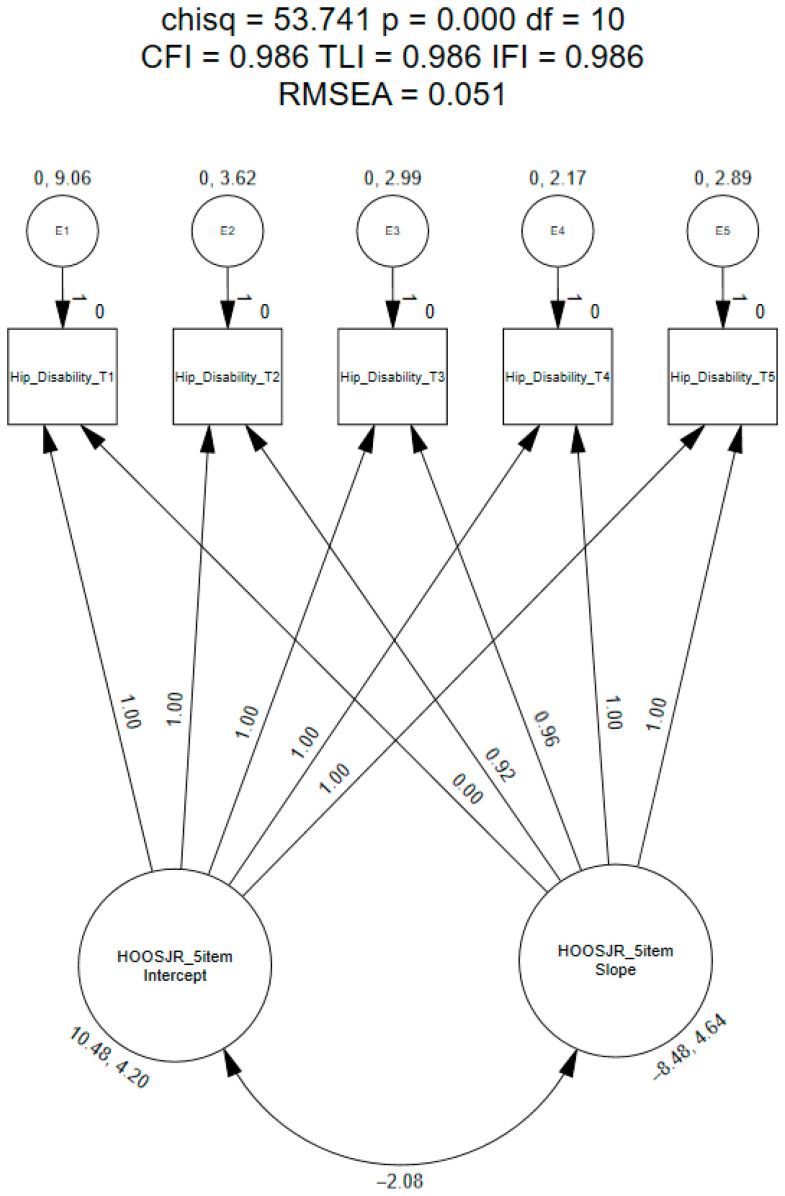
Exploratory latent growth curve model of the modified five-item HOOS-JR.

**Table 1 healthcare-12-01024-t001:** Goodness-of-fit indices for scale structure at each time point.

Modified Five-Item HOOS-JR	χ^2^	*df*	CFI	TLI	IFI	RMSEA	SRMR
Preoperative	65.33	5	0.982	0.965	0.982	0.085	0.024
6 months postoperative	59.84	5	0.982	0.964	0.982	0.081	0.025
1 year postoperative	94.98	5	0.974	0.949	0.975	0.104	0.028
2 years postoperative	138.08	5	0.964	0.929	0.965	0.126	0.035
3 years postoperative	63.30	5	0.986	0.972	0.986	0.084	0.021

**Table 2 healthcare-12-01024-t002:** Goodness-of-fit indices for longitudinal invariance across time points.

Modified Five-Item HOOS-JR	χ^2^	*df*	χ^2^_diff_ (df_diff_)	CFI	CFI_diff_	TLI	RMSEA	SRMR
Preoperative	65.33	5	-	0.982	-	0.965	0.085	0.024
6 months postoperative	59.84	5	-	0.982	-	0.964	0.081	0.025
1 year postoperative	94.98	5	-	0.974	-	0.949	0.104	0.028
2 years postoperative	138.08	5	-	0.964	-	0.929	0.126	0.035
3 years postoperative	63.30	5	-	0.986	-	0.972	0.084	0.021
Configural (equal form)	804.79	216	-	0.975	-	0.965	0.040	0.028
Metric (equal loadings)	950.73	232	145.94 (16)	0.969	0.006	0.960	0.043	0.033
Equal factor variances *	1091.45	236	**286.66 (20)**	0.964	**0.011**	0.952	0.047	0.059
Scalar (equal indicator intercepts)	1507.46	248	**702.67 (32)**	0.946	**0.012**	0.935	0.055	0.033
Equal latent means *	4128.12	252	**3323.33 (36)**	0.835	**0.140**	0.804	0.096	**0.232**

***** = Substantive questions; **Bolded** = did not meet cut-off criteria.

**Table 3 healthcare-12-01024-t003:** Goodness-of-fit indices for partial longitudinal invariance analyses across time points.

Modified Five-Item HOOS-JR	χ^2^	*df*	χ^2^_diff_ (df_diff_)	CFI	CFI_diff_	TLI	RMSEA	SRMR
Preoperative	65.33	5	-	0.982	-	0.965	0.085	0.024
6 months postoperative	59.84	5	-	0.982	-	0.964	0.081	0.025
1 year postoperative	94.98	5	-	0.974	-	0.949	0.104	0.028
2 years postoperative	138.08	5	-	0.964	-	0.929	0.126	0.035
3 years postoperative	63.30	5	-	0.986	-	0.972	0.084	0.021
Configural (equal form)	804.79	216	-	0.975	-	0.965	0.040	0.028
Metric (equal loadings)	950.73	232	145.94 (16)	0.969	0.006	0.960	0.043	0.033
Equal factor variances *	1091.45	236	**286.66 (20)**	0.964	**0.011**	0.952	0.047	0.059
Scalar (equal indicator intercepts) **	1073.83	244	269.04 (28)	0.965	0.010	0.957	0.040	0.031
Equal latent means *	3962.49	248	**3157.70 (32)**	0.842	**0.133**	0.809	0.095	**0.262**

***** = Substantive questions; **Bolded** = did not meet cuff off criteria; ** = release of item 5 intercept at level of invariance testing.

## Data Availability

The datasets analyzed during the study are not publicly available per study protocol. De-identified data may be available from the corresponding author with permission from the Cedar-Sinai Office of Research Compliance and Quality Improvement, the Kerlan-Jobe Institute, and the University of Idaho upon reasonable request.

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
