# Peer review of "Longitudinal Analysis and Latent Growth Modeling of the Modified Hip Dysfunction and Osteoarthritis Outcome Score for Joint Replacement (HOOS-JR)"

_healthcare, 2024, doi:10.3390/healthcare12101024_

Round 1

Reviewer 1 Report

Comments and Suggestions for Authors

Congratulations to the authors. The topic is good and the article is well written. The procedures are well explained, especially in the method and discussion section. I suggest the following minor corrections for authors.

Give numerical values of the results in the abstract.

-How was the sample group reached and what sampling method was used? It should be explained.

-Please explain the data collection process in detail. Where, when, at what time of day and by whom were the data collected? Please explain this section in detail.

-How was the sample group reached and what sampling method was used? It should be explained.

-Section 2.2. Data Source and Participants, I did not see any explanations about participants. Please add it.

It would be better to move the participant numbers mentioned in the results section of the study to the method section.

-I think that please add the practical applications section study.

Author Response

Dear Reviewer 1,

Congratulations to the authors. The topic is good and the article is well written. The procedures are well explained, especially in the method and discussion section. I suggest the following minor corrections for authors.

Reply: Thank you so very much for taking time to review our manuscript and provide feedback to improve our manuscript. Please see the responses to your feedback below.

  1. Give numerical values of the results in the abstract.

Reply: Thank you for your comment. We have added numerical values to the results section of the abstract. The results section from lines 23-28 now reads “The resulting CFA met most of the goodness-of-fit indices (CFI = 0.964-0.982; IFI = 0.965-0.986; SRMR = 0.021-0.035). Longitudinal analysis did not meet full invariance, exceeding the scalar invariance model (CFIDIFF = 0.012; χ2DIFF test = 702.67). Partial invariance requirements were met upon release of the intercept constraint associated with item five (CFIDIFF test = 0.010; χ2DIFF = 1073.83). Equal means model did not pass recommended goodness-of-fit indices (CFIDIFF = 0.133; χ2DIFF 3962.49).”

  1. Please explain the data collection process in detail. Where, when, at what time of day and by whom were the data collected? Please explain this section in detail.

Reply: Thank you for your comment. However, the methodology of this study is listed from lines 141-149. We stated that the data base was queried for THA procedures between 2014-2020 from the SOS database. The SOS database is an international database where multiple surgeons from different facilities which is outlined in the manuscript as well. Additionally, we sought to keep this explanation similar to other previously published manuscripts:

  1. Allred C, Reeves AJ, Casanova MP, Cady AC, Baker RT. Multi-group invariance testing of the knee injury osteoarthritis outcome score for joint replacement scale. Osteoarthritis and Cartilage Open. 2022 Dec 1;4(4):100296.
  2. Dluzniewski A, Allred C, Casanova MP, Moore JD, Cady AC, Baker RT. Longitudinal Invariance Testing Of The Knee Injury Osteoarthritis Outcome Score For Joint Replacement Scale (KOOS-JR). International Journal of Sports Physical Therapy. 2023;18(5):1094.
  3. Quintana DT, Casanova MP, Cady AC, Baker RT. Assessing the Structural Validity of the Knee Injury and Osteoarthritis Outcome Score Scale. InHealthcare 2024 Feb 6 (Vol. 12, No. 4, p. 414). MDPI.
  4. Miley EN, Pickering MA, Cheatham SW, Larkins L, Cady AC, Baker RT. Psychometric analysis of the Hip Disability and Osteoarthritis Outcome Score Joint Replacement (HOOS-JR). Osteoarthritis and Cartilage Open. 2024 Mar 1;6(1):100435.

  1. How was the sample group reached and what sampling method was used? It should be explained.

Reply: Thank you for posing this comment. Of note, we queried the SOS database for all patients who completed the HOOS-JR that were enrolled in the database. This information is explained between lines 141-149.

  1. Section 2.2. Data Source and Participants, I did not see any explanations about participants. Please add it.

Reply: Thank you for your comment. Section 2.1 is the data source and participants; however, I accidentally labeled 2.2 incorrectly. As such, this has been updated to now be 2.2. Hip Disability and Osteoarthritis Outcome Score Joint Replacement.

  1. It would be better to move the participant numbers mentioned in the results section of the study to the method section.

Reply: Thank you for your comment. However, we seek to keep the format of this manuscript similar in comparison to other previously published manuscripts. Please see below the following citations below for similar resulting manuscripts:

  1. Allred C, Reeves AJ, Casanova MP, Cady AC, Baker RT. Multi-group invariance testing of the knee injury osteoarthritis outcome score for joint replacement scale. Osteoarthritis and Cartilage Open. 2022 Dec 1;4(4):100296.
  2. Dluzniewski A, Allred C, Casanova MP, Moore JD, Cady AC, Baker RT. Longitudinal Invariance Testing Of The Knee Injury Osteoarthritis Outcome Score For Joint Replacement Scale (KOOS-JR). International Journal of Sports Physical Therapy. 2023;18(5):1094.
  3. Quintana DT, Casanova MP, Cady AC, Baker RT. Assessing the Structural Validity of the Knee Injury and Osteoarthritis Outcome Score Scale. InHealthcare 2024 Feb 6 (Vol. 12, No. 4, p. 414). MDPI.
  4. Miley EN, Pickering MA, Cheatham SW, Larkins L, Cady AC, Baker RT. Psychometric analysis of the Hip Disability and Osteoarthritis Outcome Score Joint Replacement (HOOS-JR). Osteoarthritis and Cartilage Open. 2024 Mar 1;6(1):100435.

  1. I think that please add the practical applications section study.

Reply: Thank you for your comment. Currently, most of the practical applications are embedded within the discussion section. Please see the following two examples of our practical applications discussed.

Lines 422-427, we state “When the means were not constrained to be equal, patients responded with the highest score (i.e., more hip disability) preoperatively, and the lowest scores (i.e., less hip disability) at 2- and 3-years postoperative. However, the majority of improvement in scores happened between preoperative and 6-months postoperatively; these findings are similar to previous researchers who noted the most improvement in HOOS-JR scores within the first 6-months postoperative THA [39,40,51,52].”

Another example, lines 488-492, we state “Our findings reveal that those who are younger reported greater improvement in their hip disability from preoperative to 3-years postoperatively compared to other age groups [43]. Our findings support prior research that demonstrates younger patients report faster recovery patterns following a THA when compared to older patients [40].”

The way that we have embedded practical applications into our manuscript is similar to another previous published manuscript within the same journal (see below for the citation). Please note, however, if I am mis-understanding your comment, please clarify and I can update accordingly. Thank you again for your comment and let me know how I can better update the manuscript.

  1. Quintana DT, Casanova MP, Cady AC, Baker RT. Assessing the Structural Validity of the Knee Injury and Osteoarthritis Outcome Score Scale. InHealthcare 2024 Feb 6 (Vol. 12, No. 4, p. 414). MDPI.

Reviewer 2 Report

Comments and Suggestions for Authors

- In the first paragraph of the introduction, I recommend that the authors add, 'There are other causes for THA, such as DDH and AVN, in those under 65 years of age.' This could be misinterpreted as OA in 50-64 year olds.

- The author reports 'age group' from the beginning of the paper, but detailed subgroup information (e.g., under 45, 45-54, 55-64, 65-74, 75+) appears first on line 476. Correct so that detailed age group information is available starting with the first 'Age Group.'

- Add a reference to lines 489 through 499.

- Include in the limitations of the study the limitations of postoperative care (rehabilitation). It is unfortunate that the study does not analyze the surgical approach.

Author Response

Dear Reviewer 2,

  1. In the first paragraph of the introduction, I recommend that the authors add, 'There are other causes for THA, such as DDH and AVN, in those under 65 years of age.' This could be misinterpreted as OA in 50-64 year olds.

Reply: Thank you for your comment. I have added the following sentence to the introduction from lines 44-46. However, of note: OA is becoming much more prevalent in the 50-64 year group as well in the US. “In addition to arthritis, other causes for patients undergoing a THA include developmental dysplasia of the hip and avascular necrosis, in patients younger than 65.”

  1. The author reports 'age group' from the beginning of the paper, but detailed subgroup information (e.g., under 45, 45-54, 55-64, 65-74, 75+) appears first on line 476. Correct so that detailed age group information is available starting with the first 'Age Group.'

Reply: Thank you for your comment. I have updated the methods of the study discussing multigroup invariance testing on line 234 to now state “characteristics including age groups [i.e. < 45, 45-54, 55-64, 65-74, ≥ 75) and sex (i.e., male, female).”

  1. Add a reference to lines 489 through 499.

Reply: Thank you for your comment. I have added a few references to lines 489-499 to support the conclusions drawn from the findings. More specifically, I have added references 35 and 48.

  1. Include in the limitations of the study the limitations of postoperative care (rehabilitation). It is unfortunate that the study does not analyze the surgical approach.

Reply: Thank you for your comment. We added the following update to line 516 “In addition, clinicians should practice caution using the 5-item HOOS-JR in clinical practice and throughout postoperative care when examining group differences in scores of the modified 5-item HOOS-JR in subgroups not yet analyzed.”

In addition, we agree it is unfortunate that the study does not assess PRO score differences across patients of different surgical approaches. Hopefully, future research could assess these differences using multigroup invariance testing.

Reviewer 3 Report

Comments and Suggestions for Authors

This is an excellent manuscript and can be considered for publication in the Healthcare journal. The study was a continuation of a previous study by the same authors. It is worth to highlight that the results are promising and give new insight for future studies in the field of orthopedics implants. There are several comments to be addressed before can be accepted.

1. The research gap is unclear. It is better to revise again the introduction part to make sure readers understand the importance of this study to be conducted based on the previous studies (from literature and author's previous studies)

2. It is better to include more citation that can support the results and discussion.

Author Response

Dear Reviewer 3,

This is an excellent manuscript and can be considered for publication in the Healthcare journal. The study was a continuation of a previous study by the same authors. It is worth to highlight that the results are promising and give new insight for future studies in the field of orthopedics implants. There are several comments to be addressed before can be accepted.

Reply: Thank you so very much for the time and dedication you spent providing feedback to improve our manuscript. Please see our responses to your feedback below.

  1. The research gap is unclear. It is better to revise again the introduction part to make sure readers understand the importance of this study to be conducted based on the previous studies (from literature and author's previous studies)

Reply: Thank you for your feedback. We have removed some content from the introduction that we feel may limit the impact of the literature review outlined. Additionally, we have added the following two sentences at the beginning of the last paragraph to summarize the need for the study pertaining to the gap in literature. Lines 130-133 now read “ As studies have failed to support the use of the 6-item HOOS-JR and identified a new 5-item HOOS-JR, longitudinal assessment of the proposed scale is warranted. In addition, studies assessing LCG modeling or assessing if differences exist based on age and sex of any forms of the HOOS-JR are lacking in the literature.”

  1. It is better to include more citation that can support the results and discussion.

Reply: Thank you very much for your feedback. We added a few more citations to the results in lines 262, 265, 274, 275, 279, 281, 298, 301, 321, 344, and 346. In addition, we have added a few more citations to support the discussion section throughout lines 379, 420-422, 498-504.

Reviewer 4 Report

Comments and Suggestions for Authors

Abstract :

- Delete terms: "Background, Methods, Results, Conclusions.". The abstract should be in one long, single paragraph (refer to the author's instructions).

Introduction:

The introduction is a bit too long, with details that could be integrated into the discussion. Please try to make it more concise.

References :

Ref. n°6, 37, and 49: You should include the journal with abbreviations, as stated in the journal authors' instructions.

Author Response

Dear Reviewer 4,

  1. Abstract : Delete terms: "Background, Methods, Results, Conclusions.". The abstract should be in one long, single paragraph (refer to the author's instructions).

Reply: Thank you for your comment. I have deleted the headings in the abstract.

  1. Introduction: The introduction is a bit too long, with details that could be integrated into the discussion. Please try to make it more concise.

Reply: Thank you for your feedback. We have removed some content from the introduction that we feel may limit the impact of the literature review outlined. We believe it is important to outline previous psychometric work in the literature to help set up the outlined gap in the literature and the purpose of this study. However, if there are areas that you still believe warrant removal, we are open to specific recommendations of reorganization. Hopefully, however, the updates we made will address your feedback.

  1. References : n°6, 37, and 49: You should include the journal with abbreviations, as stated in the journal authors' instructions.

Reply: Thank you for your feedback. I have updated Reference number 6 with the correct journal abbreviation. However, numbers 37 and 49 do not have an official journal abbreviation according to NLM Catalog.
